# Melanoma Detection by AFM Indentation of Histological Specimens

**DOI:** 10.3390/diagnostics12071736

**Published:** 2022-07-17

**Authors:** Byoungjun Jeon, Hyo Gi Jung, Sang Won Lee, Gyudo Lee, Jung Hee Shim, Mi Ok Kim, Byung Jun Kim, Sang-Hyon Kim, Hyungbeen Lee, Sang Woo Lee, Dae Sung Yoon, Seong Jin Jo, Tae Hyun Choi, Wonseok Lee

**Affiliations:** 1Interdisciplinary Program for Bioengineering, Graduate School, Seoul National University, Seoul 08826, Korea; bjjeon125@gmail.com; 2School of Biomedical Engineering, Korea University, Seoul 02841, Korea; paul02144@korea.ac.kr (H.G.J.); skylsw5@gmail.com (S.W.L.); dsyoon@korea.ac.kr (D.S.Y.); 3Interdisciplinary Program in Precision Public Health, Korea University, Seoul 02841, Korea; 4Department of Biotechnology and Bioinformatics, Korea University, Sejong 30019, Korea; lkd0807@korea.ac.kr; 5Interdisciplinary Graduate Program for Artificial Intelligence Smart Convergence Technology, Korea University, Sejong 30019, Korea; 6Department of Plastic and Reconstructive Surgery, Research Services, Seoul National University Bundang Hospital, Seongnam 13620, Korea; xmylife@empas.com; 7Department of Plastic and Reconstructive Surgery, Institute of Human Environment Interface Biology, Seoul National University College of Medicine, Seoul 03087, Korea; ahh000@nate.com (M.O.K.); bjkim79@gmail.com (B.J.K.); 8Department of Internal Medicine, Keimyung University Dongsan Medical Center, Daegu 41931, Korea; mdkim9111@dsmc.or.kr; 9Department of Biomedical Engineering, Yonsei University, Wonju 26493, Korea; hyungbeen.lee@gmail.com (H.L.); yusuklee@yonsei.ac.kr (S.W.L.); 10R&D Center of Curigin Ltd., Seoul 04778, Korea; 11Astrion Inc., Seoul 02841, Korea; 12Department of Dermatology, Seoul National University College of Medicine, Seoul 03087, Korea; 13Department of Electrical Engineering, Korea National University of Transportation, Chungju 27469, Korea

**Keywords:** malignant melanoma, benign nevus, atomic force microscopy, nanoindentation, mechanical characterization

## Abstract

Melanoma is visible unlike other types of cancer, but it is still challenging to diagnose correctly because of the difficulty in distinguishing between benign nevus and melanoma. We conducted a robust investigation of melanoma, identifying considerable differences in local elastic properties between nevus and melanoma tissues by using atomic force microscopy (AFM) indentation of histological specimens. Specifically, the histograms of the elastic modulus of melanoma displayed multimodal Gaussian distributions, exhibiting heterogeneous mechanical properties, in contrast with the unimodal distributions of elastic modulus in the benign nevus. We identified this notable signature was consistent regardless of blotch incidence by sex, age, anatomical site (e.g., thigh, calf, arm, eyelid, and cheek), or cancer stage (I, IV, and V). In addition, we found that the non-linearity of the force-distance curves for melanoma is increased compared to benign nevus. We believe that AFM indentation of histological specimens may technically complement conventional histopathological analysis for earlier and more precise melanoma detection.

## 1. Introduction

Melanoma originated from aberrant melanin-pigmented cells and is responsible for approximately 80% of patient deaths from skin cancer [1,2]. Because the survival of melanoma depends strongly on whether cancer invades from the epidermis to the dermis, it is important to identify early-stage melanoma before the invasion. Commonly, the pathologic examination has been performed on histological specimens to determine whether they are benign or malignant, but this method often results in both false positives and false negatives [3,4]. Accordingly, other diagnostic techniques such as optical examination [5,6,7] and ultrasound [8,9] have been developed, but these are still inaccurate as well; in particular, they suffer from low sensitivity owing to either epidermal structures such as hair or scars or motion artifacts during measurement [10].

To overcome these limitations, researchers have developed complementary methods such as image processing and impedance measurement [8,11,12,13,14,15], but demarcating a boundary between a benign nevus and melanoma is still uncertain [16]. This uncertainty is thought to derive from the anatomical structure of the epidermis, where there is no extracellular matrix (ECM), so melanoma tumorigenesis differs from that of other cancers [17,18]. Many cancers undergo ECM remodeling on progression (e.g., in stiffness and configuration) [19,20,21,22], but melanoma does not undergo this process due to the absence of the ECM. It indicates that melanoma tissue exhibits few discernable changes in morphological or physical properties compared to other cancers [23,24].

Given this consideration, we need to pay attention to the tumorigenesis that is specific to melanoma rather than other cancers, including the minute changes in the molecular interactions (e.g., melanin transfer and pigmentation). We believed that tumorigenesis would modulate the mechanical properties of melanoma tissues in the interface between the epidermis and the dermis (i.e., nearby basal layers) [1,25,26]. The changes in the mechanical properties of melanoma tissues would then derive from the cellular configuration changes within melanoma tissues in comparison with both nevus and normal tissues. However, with conventional methods, a sharp distinction between a benign nevus and malignant melanoma is still challenging [27,28]. 

Meanwhile, we noted the typical symptoms of melanoma blotches—irregular color, shape, or both—in the outermost layer of skin, whereas benign nevus revealed regular round shapes and a uniform color (Figure 1a), and this trend was consistent with a previous report that studied visual inspection [29]. Based on our visual inspection knowledge, our detailed rationale for the development of malignant melanoma with melanin transfer and pigmentation processes in the vicinity of a basal layer is as follows: (i) in the epidermis, melanoma neoplasm develops from abnormal melanin transfer and pigmentation from the surrounding keratinocytes and melanocytes [30,31]; (ii) in this process, the cell-cell junctions of keratinocytes acting as homogeneous melanin transfer pathways could be confined and thus drive the development of a benign pigmented area (i.e., a nevus) [1]; (iii) in contrast, in melanoma development, melanin transfer and pigmentation become irregular by the invasion of melanoma cells, and the cell-cell junctions between cancer and normal cells act as heterogeneous melanin transfer pathways (Figure 1b) [17,27]. Accordingly, subtle differences in neoplasm formation may lead not only to the different visual signatures of the skin blotches but also to the obvious contrast in the mechanical properties between nevus and melanoma tissues. Therefore, we hypothesized that distinct local elastic properties of both lesion tissues (benign nevus and malignant melanoma) are implicit in the biopsied specimens and can be measured (Figure 1c). Recently, He et al. found a trend that melanoma tissue has higher density and more heterogeneous distribution patterns of melanin than benign tissue using the three-dimensional imaging of the tissue with confocal photothermal microscopy [32], which may support our hypothesis.

Up to date, biopsied cancer tissues have been employed for mechanical characterization. However, it has a limitation that the mechanical properties of tissues can change over time after biopsy [33,34,35]. The histological specimens, commonly used for diagnosis in hospitals, can overcome this difficulty in handling and storing biopsied samples to study their mechanical properties [33]. These specimens can be easily handled and can be reused because they can withstand long-term storage due to fixed and mummified tissue. As such, our goals were to verify (i) whether the mechanical characterization of the histological specimen would be consistent with the results from conventional histopathological analysis and (ii) whether it could provide abundant information about benign versus malignant epidermal tissues, complementing the traditional histopathological examinations.

Atomic force microscopy (AFM) is a multi-functional device that can analyze a sample’s morphology, mechanical strength, and surface charge by its nano-scale cantilever [36,37,38,39]. In this report, we demonstrate a robust mechanical characterization of histological specimens using atomic force microscopy (AFM) indentation for discrimination among normal, benign nevus, and melanoma. Specifically, the histograms of the elastic modulus of melanoma displayed multimodal Gaussian distributions, exhibiting heterogeneous mechanical properties, in contrast with the normal and benign nevus. We also identified the mechanical characteristics of each histological specimen regardless of blotch incidence by sex, age, anatomical site (e.g., thigh, calf, arm, eyelid, and cheek), or cancer stage (I, IV, and V). Moreover, we found that the non-linearity of the measured force-distance (FD) curves for melanoma specimens tends to be increased compared to normal and benign nevus. This implies that the non-linearity of FD curves will be another significant metrics for melanoma detection in cooperation with the elastic modulus mapping. We believe that the AFM indentation of the histological specimen can be considered a useful and complementary technique not only for providing an excellent complement to histopathological examination for precise diagnosis but also for predicting the invasiveness and ablation margin during oncological surgery.

## 2. Results

### 2.1. Surface Profiling of Histological Specimens by Using AFM

Prior to our AFM indentation experiments, we performed AFM imaging for morphological analysis of histological specimens (Figure 2a–c and Appendix A). The AFM micrographs showed circular or elliptical craters (approximately 10 μm), which are presumed to be cell markers (Figure 2d–f). We found that the AFM topographies notably differed among normal, benign nevus, and melanoma. We believed that the higher density and correspondingly smaller sizes of the craters had evolved during nevus development, but there were still some aligned crater structures in the nevus specimens. In contrast, the melanoma showed irregular crater shapes, sizes, densities, and even alignments. These structural irregularities could have been formed during the transformation to malignancy [27,28], and they can affect the changes in the physical properties of the samples. 

For the quantitative analysis beyond that of appearance, we compared the histograms of height distribution from each AFM micrograph in Figure 2g–i. The standard deviations (σ) of the histograms, fitted to Gaussian distributions, are shown in Figure 2j. In addition, we attempted to calculate the surface roughness (S_q_) from the AFM micrographs, but we found that both metrics (i.e., σ and S_q_) were ineffective at discriminating among normal, benign nevus, and melanoma (Figure 2k and Appendix A). The results of the morphological analysis were similar because the histological specimens were finely sectioned at the same thickness (4 μm). Although the microscopic analysis (i.e., AFM imaging) showed higher-resolution images compared with macroscopic inspection (e.g., simple visual inspection using ABCDE criteria), we comprehensively demonstrated the vulnerability of AFM image analysis for melanoma detection using histological specimens.

### 2.2. Mechanical Properties of Histological Specimens That Can Discriminate between Benign Nevus and Melanoma

To overcome the vulnerability of AFM image analysis, we performed AFM indentation of the histological specimens in the vicinity of the epidermis (Figure 3a–c). The mechanical properties of specimens can be quantified by using FD curve measurements, which we acquired from 100 independent locations throughout the entire lesion. To confirm whether the histological specimen is softer than the substrate (i.e., glass) or not, we evaluated the mechanical properties of the glass substrate. We observed that the elastic modulus of the substrate that supported the histological specimens exceeded the usual range of the mechanical properties of histological specimens, indicating that the heterogeneous properties of the histological specimens are not influenced by the substrate (Appendix A). Therefore, we were convinced that our experimental setup and the following results are reasonable.

All the FD curves are presented from the contact point to the maximum indentation depth in the form of approach curves after the contact point. The displayed FD curves (Figure 3d–f) are a set of the approach curves where the contact points of the FD curves fit in the origin (*d* = 0 μm, F(d) = 0 μN). The majority of FD curves from the normal sample exhibited linear behavior during the indentation (Figure 3d). The majority of FD curves from the benign sample showed relatively steeper slopes in the linear region than did the normal sample, implying that the benign sample retains higher resistance to elastic deformation. It is noteworthy that a few of the FD curves in the benign sample revealed non-linear characteristics (Figure 3e). In contrast, the FD curves for the melanoma specimen exhibited non-linear characteristics (Figure 3f), suggesting that benign nevus is relatively harder than the normal sample, whereas melanoma appears to combine softness and hardness. We will discuss the non-linear characteristics of FD curves in detail later in this article.

We conducted the stiffness mapping of the specimens; stiffness maps are derived from FD curves, and they show the deformation of samples. In our study, a representative stiffness map of a benign nevus revealed that overall stiffness was higher than that in the normal sample (Figure 3g,h). In contrast, the stiffness map of the melanoma displayed randomly blended colors, indicating a mixture of soft and hard materials, unlike with normal or nevus samples (Figure 3i). This suggests that the local elastic properties of melanoma are heterogeneous.

To quantify the extent of any heterogeneity, we calculated the elastic modulus (*E_a_*) using the FD curve data. The average *E_a_* in the normal sample was 401 ± 148 MPa, and that of the benign nevus was 575 ± 107 MPa. Here, we found two things: The elastic modulus of the benign nevus was generally 100 MPa higher than those of the normal sample, and the histograms of the elastic modulus distributions of the normal and benign samples followed Gaussian distributions with single peaks (Figure 3j,k). In contrast, the elastic modulus of the melanoma was described by multiple Gaussian distributions (188 ± 78, 497 ± 110, and 787 ± 56 MPa; Figure 3l). This result implies that the epidermis of melanoma consists of heterogeneous materials including soft matter. It is surprising because such mechanical characteristics are of tumor tissue with ECM [27,28,40]. Perhaps, the role of ECM in the tumorigenic process is seemed to replace by some factors (e.g., melanin transfer and pigmentation) in melanoma development. Meanwhile, we checked whether the samples would retain similar trends in elastic properties even after long periods of time, and this is of practical clinical importance as well; our AFM indentation testing verified that the characteristics of the melanoma such as the multimodal Gaussian distribution survived for more than three months (Appendix A). Previous studies have found that using histological specimens has a strong advantage in overcoming the weakness of the time-dependent biodegradation of biopsied tissues [35,41]. 

The results for the mechanical properties of all specimens are summarized in Table 1 and shown in Appendix A. We found that the elastic modulus of all histological specimens could be separated into three segmented regions across the full range: the ranges were region I (0–300 MPa), region II (300–600 MPa), and region III (600–900 MPa). The different specimens’ *E_a_* value ranges depended on the specimens’ origins; for instance, the *E_a_*s for all normal samples were in region II (300–600 MPa), exhibiting a single-mode Gaussian distribution, and the *E_a_*s for the benign nevus spread across regions II and III and displayed single-mode Gaussian distributions. However, the elastic modulus of the melanoma exhibited multimodal Gaussian distributions that ranged from regions I to III and mainly consisted of three peaks (occasionally there were two peaks). It should be noted that this classification is highly useful in testing the versatility of our approach with all epidermal lesions regardless of sample type.

### 2.3. Versatility of the Mechanical Specimen Signatures Regardless of Age, Sex, or Site

The epidermis is the outermost layer of the skin, which suggests that it can be easily deformed and that it has different mechanical properties depending on age, sex, and site [42,43]. To clinically apply our methodology, it was essential to verify whether the mechanical signatures for discrimination would hold constant irrespective of skin tissue type. Thus, we classified the mechanical properties of the specimens according to age (1–81 years), sex (male or female), and site (thigh, inguinal, cheek, leg, arm, back, flank, abdomen, and elbow). Surprisingly, the specimen signatures varied based on normal, benign nevus, or melanoma samples regardless of age, sex, or site (Figure 4a–i). The average *E_a_*s for the normal and benign nevus samples fell in region II or regions II and III—showing single-mode Gaussian distributions—whereas those of the melanoma mostly had multiple peaks and were distributed throughout all the elastic regions (I, II, and III). In detail, the *E_a_*s for the normal samples ranged from 370 to 521 MPa, all within region II. The benign nevus had higher *E_a_*s, ranging from 441 to 848 MPa, in regions II and III. In contrast, the melanoma displayed three peak Gaussian distributions, and each peak had a different average: *E*_1*a*_ = 158–274 MPa (region I), *E*_2*a*_ = 363–542 MPa (region II), and *E*_3*a*_ = 606–893 MPa (region III).

Whether the melanoma cancer stage can distort the mechanical signatures is another important factor in determining the clinical applicability of our method. It is well-known that melanomas become darker, more distorted, and physically harder as the stage increases (Figure 4j). Accordingly, we determined the cancer stage of each melanoma based on Clark level (see Methods) and then inspected all the melanoma by AFM indentation. Notably, all samples exhibited multiple Gaussian peaks; each *E_a_* occupied its own elastic region (Figure 4k). From our results, it is plausible that the mechanical signature of the multiple peaks survives in melanoma irrespective of the cancer stage, even in stage I, and the existence of these multiple peaks in stage I melanoma provides us with a strong advantage in early detection.

At this stage, we needed to scrutinize in detail how the melanoma cancer stage affected its mechanical characteristics. One particular point is that, unlike other cancers, melanoma’s mechanical properties appeared to vary significantly depending on the stage. Specifically, each of our melanoma samples showed the most prevalent peak (*P_m_*)—one with the highest population—among its multiple peaks (Appendix A). We considered the *P_m_* and its corresponding elastic region, I, II, or III, as important parameters. We conducted our statistical analyses of each category based on the two parameters above, and we present the results in Figure 4l and Appendix A. We found no significant differences in the mechanical properties of melanoma by sex, age, or site. However, we found that the melanoma specimens tended to become harder as the cancer stage advanced. In particular, all early-stage (stage I) melanoma samples had their own *P_m_* values, which were all in the region I, which implied that the specimens contained large amounts of soft material. However, in the late melanoma stages (IV and V), the *P_m_* values ranged across all three elastic regions, I, II, and III. This finding indicates that at the late stage, hard material fills the samples in larger amounts than in the early stage. This phenomenon is consistent with the fact that melanin accumulates as the cancer stage advances [44]. Taking these findings together, although there were subtle differences by category and cancer stage, we found it surprising that all the melanoma samples still retained their mechanical signature of multiple Gaussian peaks.

### 2.4. Non-Linearity of FD Curves for Normal, Benign Nevus, and Melanoma 

Meanwhile, we paid attention that the FD curves for normal and benign nevus samples exhibited linear behavior whereas the FD curves of melanoma showed non-linear characteristics (Figure 3). To quantitatively compare the non-linear characteristics of each sample, we calculated the non-linearity from FD curves as follows. As shown in Figure 5a, we draw a straight line (red-dashed) between the contact point (X_min_, Y_min_) and endpoint (X_max_, Y_max_) in the FD curve. We define this straight line as the ideal curve (i.e., perfectly linear line): (1)$$\mathrm{Ideal}\;\mathrm{curve}:{\;y}_{\mathrm{linear}}=\frac{Y_{\max}-Y_{\min}}{X_{\max}-X_{\min}}\left(x-x_{\min}\right)+Y_{\min}$$

To calculate the non-linearity of FD curves, we measured the deviation between the linear line (i.e., ideal curve) and the FD curve (i.e., experimental data);
(2)$$\mathrm{Deviation}\;\left(D,\%\right)=\frac{\left|y_{\mathrm{ideal}}-y_{\mathrm{FD}}\right|}{y_{\mathrm{ideal}}}\times 100$$
where, y_ideal_ and y_FD_ are y-intercepts of the ideal and experimental data from a FD curve, respectively. We averaged the D in all data points (i.e., sampling number) of the FD curve to calculate the representative non-linearity (*NL*) characteristics for the FD curve.
(3)$$NL=\frac{1}{n}\overset{n}{\underset{i=1}{\sum }}D_{i}$$

For a certain specimen, hundreds of FD curves are existed, whereby hundreds of *NL* can be calculated. We compared the histograms of *NL* distribution from each specimen as shown in Figure 5b. Interestingly, we found that the *NL* histograms for normal and benign nevus showed a distribution of less than 40%, whereas the distribution of *NL* histograms for melanoma spreads more than 40%. For more quantitative analysis, we fit the *NL* histograms with the inverse Gaussian distributions given by;
(4)$$\sqrt{\frac{\lambda }{2{\pi x}^{3}}}\exp\left\{-\frac{\lambda }{2\mu^{2}x}{\left(x-\mu \right)}^{2}\right\}$$
where μ and λ are the mean value and the shape parameter of inverse Gaussian distribution. The values of μ were 6.18 ± 1.55, 6.40 ± 1.71, and 8.39 ± 1.38 for normal, nevus, and melanoma, respectively. It shows that the mean value of histograms is increased from normal to benign nevus to melanoma (Figure 5c), which is attributed to enlarged non-linear characteristics. By contrast, the values of λ were decreased: 5.76 ± 3.06, 5.28 ± 2.15, and 4.80 ± 1.94 for normal, nevus, and melanoma, respectively. Using the value of μ and λ, the standard deviation and the skewness can be calculated through the following two equations:(5)$$\mathrm{Standard}\;\mathrm{deviation}\;\left(\sigma \right)=\sqrt{\frac{\mu^{3}}{\lambda }}$$
(6)$$\mathrm{Skewness}=3\sqrt{\frac{\mu }{\lambda }}$$

The values of standard deviation (σ) were 61.35 ± 46.03, 54.59 ± 26.47, and 170.04 ± 134.50 for normal, nevus, and melanoma, respectively (Figure 5d). The standard deviation of the *NL* histogram for melanoma was more than twice those of the other two cases (i.e., normal and benign nevus). The values of skewness also increased from normal to benign to melanoma; 2.90 ± 0.15, 3.46 ± 0.59, and 4.26 ± 1.31, respectively (Figure 5e). Together, in melanoma development, all factors for non-linear characteristics (i.e., the values of mean, standard deviation, and skewness of *NL* histograms) were found to be increased compared to the other conditions. It is fair to say that the non-linearity of FD curves can be used as a complementary indicator of discrimination among normal, benign, and melanoma.

## 3. Discussion

Using AFM indentation, we characterized the mechanical properties of histological specimens for melanoma detection and discrimination between benign nevus and melanoma. We conclude that the mechanical signatures (the elastic region of *E_a_* and the existence of multiple peaks) for discrimination are universally acceptable regardless of age, sex, or site and that they can specifically demarcate benign nevus versus malignant melanoma samples. In addition, we discover that the non-linearity characteristics of FD curves can be used as a complementary index in collaboration with the elastic modulus characteristics. From our results, we believe that the AFM indentation of histological specimens provides an excellent complement to histopathological examination for precise diagnosis. Additionally, our method can also be applied as a promising technology not only to determine the safety margin before surgical excision for diagnosis but also to observe the invasiveness during oncological surgery. In the benign nevus, the lesion is relatively easy to be removed without a safety margin but melanoma including both lesion and its borders must be removed for preventing the recurrence or invasiveness of cancer. The accurate diagnosis of the biopsied tissue is expected to reduce the side effects from surgical excisions, such as cancer recurrence, recovery delay, and large surgery scars. Meanwhile, because invasiveness is an important factor in preventing recurrence, it is determined by using the frozen section method during oncological surgery. Although this method is much easier and faster than conventional histopathology (around 10 min vs. several hours), providing some uncertainty in identifying the invasiveness of cancer. Therefore, our technique can be considered a useful and complementary technique for predicting the invasiveness and ablation margin during oncological surgery.

## 4. Methods

### 4.1. Biopsy from Patients

All samples were collected from patients who need pathological evaluation at the Seoul National University Hospital, Seoul, Korea. A total of 33 human biopsies of tissue samples were obtained from both males and females aged 1 to 81:7 normal skin samples, 13 benign nevus samples, and 13 melanoma samples. The biopsies of each patient were evaluated by standard pathological procedures before conducting the AFM analysis, and the pathologists determined that among the 13 melanoma tissue samples, three were Clark level I, five were level IV, and the remainder were level V. Remaining sample blocks were then prepared for AFM analysis. Ethical approval was obtained from the Institutional Review Board of Seoul National University Hospital (IRB Approval Number: 1505-092-673). All experiments dealing with human or human products were conducted with informed consent and carried out in accordance with the relevant guidelines and regulations. The biopsies of each patient were evaluated by standard pathological procedures before we conducted the AFM analysis, and the pathologists determined that among the 13 melanoma tissue samples, three were Clark level I, five were level IV, and the remainder were level V. All experiments were performed by relevant guidelines and regulations. 

### 4.2. Preparation of Histological Specimens

The acquired tissue samples were formalin-fixed and paraffin-embedded according to standard histological procedures. The prepared paraffin-embedded blocks were sectioned at roughly 4 µm thickness on a microtome and transferred onto glass slides suitable for immunohistochemistry. The first sections were stained with hematoxylin and eosin (H&E) and used for traditional histopathological examination using an upright light microscope (Leica Microsystems GmbH, Wetzlar, Germany) to determine the lesion. The second sections were also stained with H&E, but they were not mounted with a cover glass; the uncovered samples were used for AFM indentation after drying.

### 4.3. Histological Analysis

We determined the histologic lesions of benign nevus and melanomas where AFM topography was obtained, and AFM indentation was measured. Specifically, we determined that the benign nevus lesions were nests of melanocytes in the lower aspect of the epidermis in the H&E sections; the melanocytes were polygonal and epithelioid with a uniform round to oval small nucleoli and clear to pale staining cytoplasm containing evenly distributed melanin granules. In contrast, we determined the melanoma lesions to be areas where atypical melanocytes were scattered or formed clusters. These atypical melanocytes contained pleomorphic, angular, and hyperchromatic nuclei and showed conspicuous cytoplasmic fixation retraction artifacts. Pigmentation was often abundant, and multinucleated tumor cells were commonly seen.

### 4.4. AFM Topography of Histological Specimens

All histological specimens which were mummified samples after fixation of tissue were stored in a desiccator. The humidity (47%) of the laboratory was constantly maintained during all AFM indentation experiments. We acquired the AFM images for topological analysis of the histological specimens using commercial AFM (XE-Bio, Park Systems, Suwon, Korea) operated in an air-conditioned environment; the image sizes were 60 × 60, 70 × 70, and 90 × 90 μm^2^, and the scan rate was 0.5 Hz. Before the AFM imaging, we moved the cantilever above the tissue lesion using an optical microscope. We obtained and analyzed all AFM images and surface roughness data using the commercial Park Systems software, XEI version 4.3.0.

### 4.5. AFM Indentation of Histological Specimens 

To investigate the mechanical properties of the histological specimens, we conducted the AFM indentation in air conditioning. For all of the indentation experiments, we used aluminum-coated cantilevers (PPP-NCHR, tip radius of curvature < 10 nm, resonant frequency in the air: 330 kHz) with spring constants of 38.70 ± 1.13 N/m and force sensitivity of 73.02 ± 0.83 V/μm. Because PPP-NCHR was adaptable to measure samples with an elastic modulus of under gigapascals [45,46]. In all the indentation experiments, the cantilevers with the spring constant and force sensitivity were carefully chosen and used for measurement. For the tissue specimens, we conducted the AFM indentations in force-volume mode, wherein we collected an array (10 × 10 points) of FD curves over the entire scan area (30 × 30 and 45 × 45 μm^2^, half of the full AFM image size). We acquired force–volume maps spaced 3 and 4.5 μm apart in a systematic manner across the entire mapping surface (Figure 3g–i). In our experiment, each FD curve consisted of 512 data points. The cantilever was brought to the specimens with the constant speed of 1 μm/s, and it was held on the tissue surface at a constant force of 2–3.8 μN depending on the mechanical differences within the histological specimens. Note that the histological specimen could withstand the micro-newton force because it was a fixed tissue with a chemical reagent rather than raw biological tissue. After the indentation experiment, we confirmed that no dramatic damage to the histological sample was observed using the charge-coupled device. We calculated the elastic modulus of each specimen through the measured FD curves, which we derived from the Hertz model provided in XEI [47].

### 4.6. Statistics 

To investigate the statistical differences in our data, peaks of histograms were analyzed by the peak analyzer application of OriginPro 2016. The statistical significance of differences between peak values in histogram was assessed with the ANOVA in OriginPro, where significance was taken as *p* < 0.05.

## Figures and Tables

**Figure 1 diagnostics-12-01736-f001:**
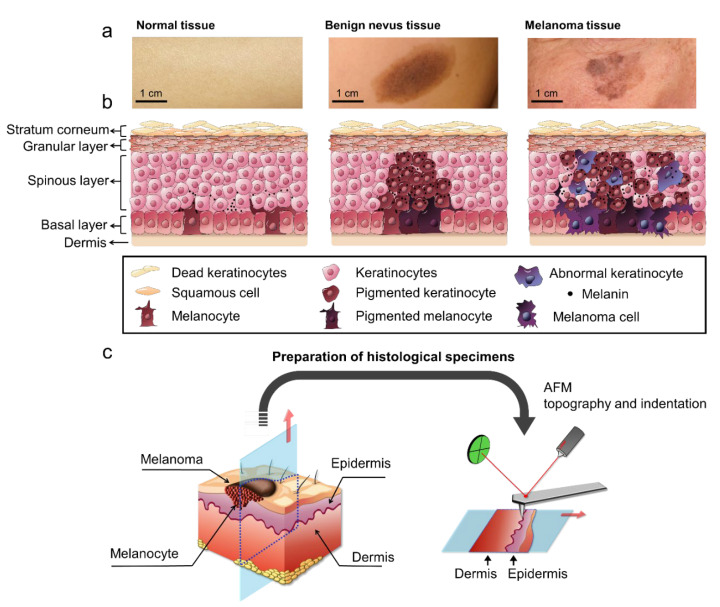
The morphological spectrum and AFM-based analysis of histological specimens. (**a**) Clinical images of the outermost layer of skin (left to right: normal, benign nevus, melanoma). (**b**) Schematic illustration of anatomical structures for each type of skin tissue and a rationale of malignant melanoma development with melanin transfer and pigmentation processes in the vicinity of a basal layer. (**c**) Schematic illustration of tissue biopsy (sampling) and AFM-based analysis.

**Figure 2 diagnostics-12-01736-f002:**
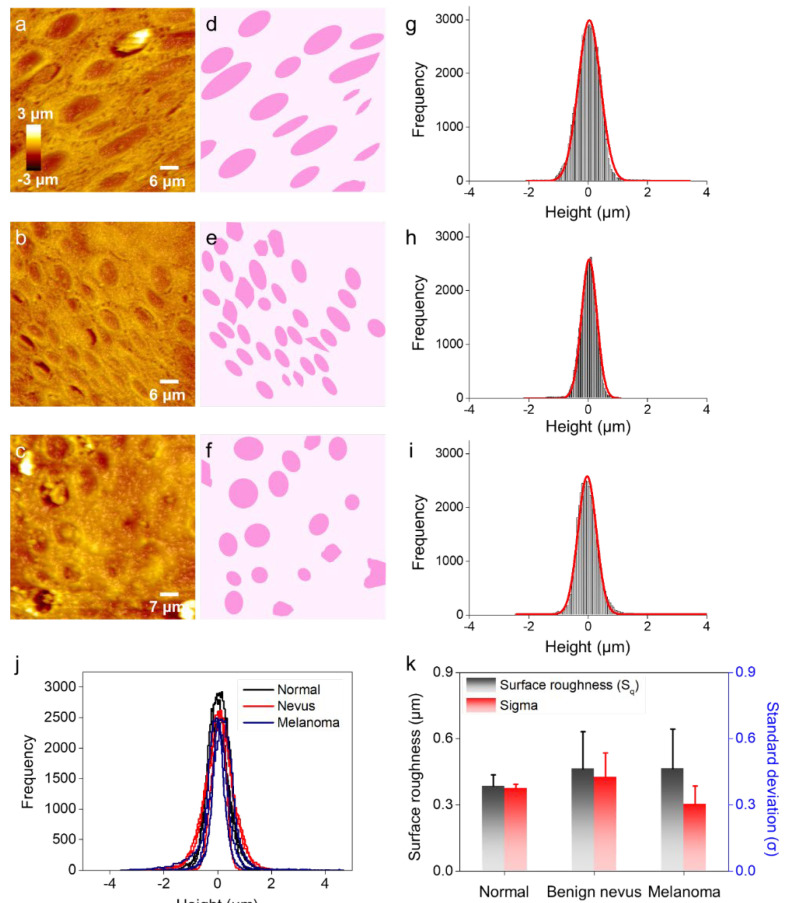
AFM images of (**a**) normal, (**b**) benign nevus, and (**c**) melanoma tissue. (**d**–**f**) An outlined cell mark was obtained from the corresponding AFM images. Statistical height histogram of the AFM images of (**g**) normal (mean ± standard deviation = 0.03 ± 0.37 μm), (**h**) benign nevus (0.03 ± 0.25 μm), and (**i**) melanoma tissue (−0.03 ± 0.31 μm). (**j**) All height histograms of the normal (solid black line), benign nevus (solid red line), and melanoma (solid blue line) specimens. (**k**) Mean surface roughness of tissue specimens extracted from AFM image processing program and standard deviation from the Gaussian fitting curve of the height histogram (**j**).

**Figure 3 diagnostics-12-01736-f003:**
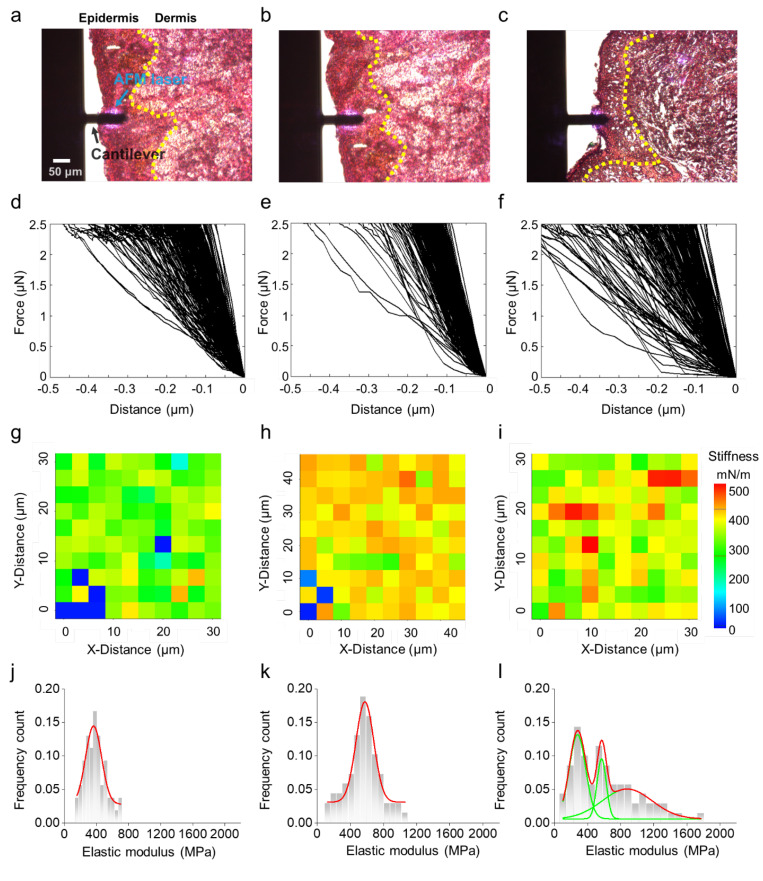
Optical images of the AFM cantilever moved above the tissue on (**a**) normal, (**b**) benign nevus, and (**c**) melanoma specimens. The yellow dotted line indicates a basal layer between the epidermis and dermis. (**d**–**f**) FD curve and representative stiffness maps (10 × 10 points) across the (**g**) normal, (**h**) benign nevus, and (**i**) melanoma specimens. (**j**–**l**) Corresponding histogram of elastic modulus distribution, which was calculated from the FD curve data.

**Figure 4 diagnostics-12-01736-f004:**
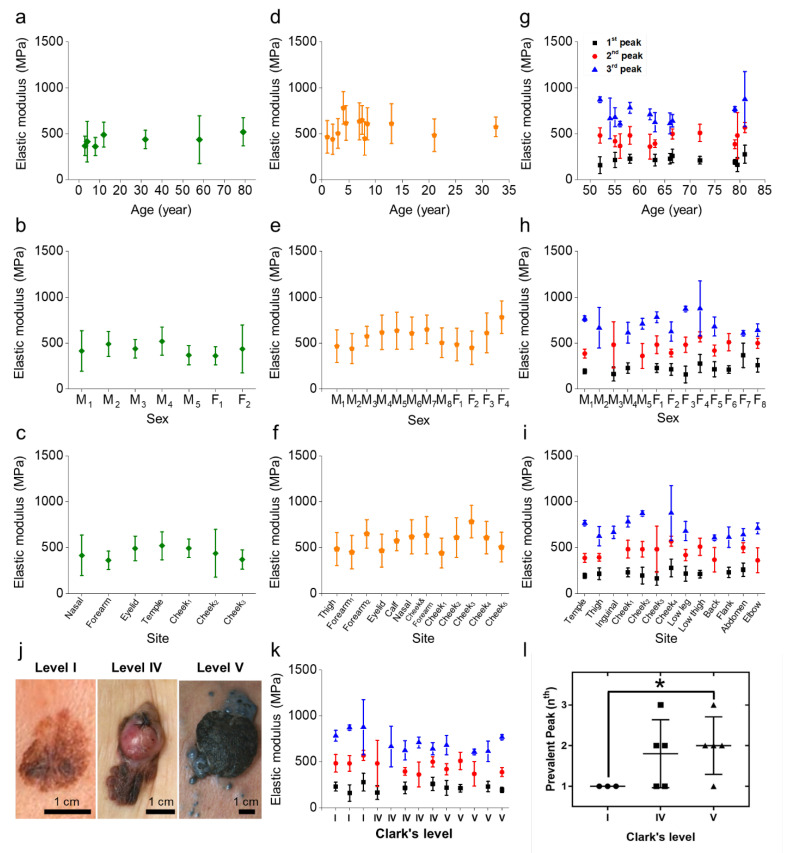
The elastic modulus of (**a**–**c**) normal, (**d**–**f**) benign nevus, and (**g**–**i**) melanoma specimens are classified by age, sex, and site. (**j**) Macroscopic images of the different stages of melanoma skin denoted by Clark level. (**k**) Means and standard deviations for the elastic modulus of the melanoma tissue specimens by cancer stage. (**l**) Plot of the prevalent peak range (nth-peak among the first to third peak regions) in elastic modulus distribution from the histograms (Appendix A), displaying the melanoma development from I to V (** p* < 0.05).

**Figure 5 diagnostics-12-01736-f005:**
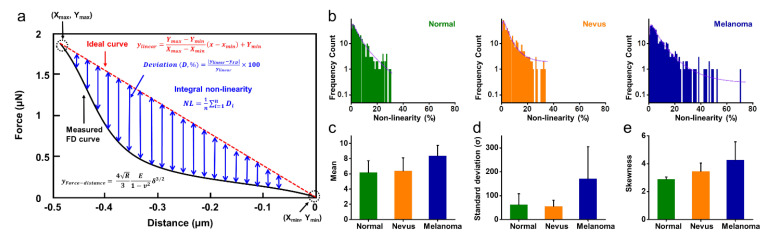
Non-linearity of FD curves of normal, benign nevus, and melanoma specimens. (**a**) Schematic illustration of the calculation of non-linearities. (**b**) The histograms of non-linearity of normal, benign nevus, and melanoma specimens were calculated from individual FD curves. The value of (**c**) mean, (**d**) standard deviation, and (**e**) skewness extracted from a model with the inverse Gaussian distribution of the histograms (**b**).

**Table 1 diagnostics-12-01736-t001:** Summary of mechanical analysis and histological examination of biopsied samples with patient information. The corresponding elastic modulus distributions are shown in Appendix A.

Case No.	Age/Sex	Biomechanical Property			Histopathological Diagnosis
		1st Peak (0~300 MPa)	2nd Peak (300~600 MPa)	3rd Peak (600~900 MPa)	
1	4/Male		415 ± 221 MPa		Normal
2	8/Female		401 ± 148 MPa		Normal
3	12/Male		491 ± 135 MPa		Normal
4	32/Male		439 ± 101 MPa		Normal
5	79/Male		521 ± 153 MPa		Normal
6	58/Female		437 ± 261 MPa		Normal
7	3/Male		370 ± 105 MPa		Normal
8	21/Female		485 ± 179 MPa		Intradermal nevus (rt. Thigh)
9	8/Female		450 ± 182 MPa		Congenital melanocytic nevus (lt. Forearm)
10	1/Male		467 ± 178 MPa		Congenital melanocytic nevus (rt. Eyelid)
11	2/Male		441 ± 164 MPa		Congenital melanocytic nevus (rt. Cheek)
12	32/Male		575 ± 107 MPa		Compound nevus (rt. Calf)
13	3/Male		505 ± 162 MPa		Congenital melanocytic nevus (rt. Cheek)
14	13/Female			611 ± 217 MPa	Compound nevus (lt. Cheek)
15	4/Female			848 ± 299 MPa	Congenital melanocytic nevus (rt. Cheek)
16	4/Male			618 ± 187 MPa	Congenital melanocytic nevus (rt. Nasal area)
17	7/Male			636 ± 203 MPa	Congenital melanocytic nevus (rt. Cheek & forearm)
18	8/Male			609 ± 176 MPa	Compound nevus (rt. Cheek)
19	7/Male			650 ± 156 MPa	Congenital melanocytic nevus (rt. Forearm)
20	12/Female			783 ± 177 MPa	Congenital melanocytic nevus (lt. cheek)
21	58/Female	229 ± 49 MPa	483 ± 97 MPa	784 ± 59 MPa	Malignant melanoma, Clark’s level I (rt. Cheek)
22	52/Female	159 ± 90 MPa	482 ± 84 MPa	873 ± 30 MPa	Malignant melanoma, Clark’s level I (lt. Cheek)
23	81/Female	278 ± 98 MPa	569 ± 55 MPa	878 ± 300 MPa	Malignant melanoma, Clark’s level I (lt. Cheek)
24	79/Male	164 ± 76 MPa	482 ± 255 MPa		Malignant melanoma, Clark’s level IV (lt. Cheek)
25	54/Male			667 ± 221 MPa	Metastatic malignant melanoma, Clark’s level IV (lt. Inguinal area)
26	63/Female	215 ± 64 MPa	394 ± 41 MPa	625 ± 105 MPa	Malignant melanoma, Clark’s level IV (rt. Thigh)
27	62/Male		361 ± 136 MPa	711 ± 59 MPa	Malignant melanoma, Clark’s level IV (lt. Elbow)
28	66/Female	259 ± 73 MPa	499 ± 56 MPa	642 ± 67 MPa	Malignant melanoma, Clark’s level IV (rt. Abdomen)
29	55/Female	216 ± 81 MPa	419 ± 61 MPa	681 ± 104 MPa	Malignant melanoma, Clark’s level V (lt. Low leg)
30	72/Female	212 ± 42 MPa	510 ± 94 MPa		Malignant melanoma, Clark’s level V (lt. Low thigh)
31	56/Female		368 ± 133 MPa	608 ± 30 MPa	Malignant melanoma, Clark’s level V (lt. Back)
32	66/Male	230 ± 56 MPa		614 ± 112 MPa	Malignant melanoma, Clark’s level V (rt. Flank)
33	79/Male	193 ± 27 MPa	387 ± 49 MPa	768 ± 28 MPa	Malignant melanoma, Clark’s level V (rt. Temple)

## Data Availability

Not applicable.

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
