# Peer review of "Melanoma Detection by AFM Indentation of Histological Specimens"

_diagnostics, 2022, doi:10.3390/diagnostics12071736_

Round 1
Reviewer 1 Report
The present manuscript tries to compare the elasticity of normal, benign and malignant skin tissue targeting the multi-gaussian distribution as a biomarker for malignancy. The idea is interesting, however there are several questions which are not addressed in the manuscript.
Questions and comments:
1. The authors work with fixed tissue preparations an even check their shelf life which is seems to last for several months. How is related the observed elasticity distributions on fixed samples to not-fixed ones? Fixation induces serious alterations, but at least the qualitative comparison would be good for the manuscript. Furhtermore, the Preparation section states that AFM measurements were effectuated on dried samples, so how would this influence the translation of the results to original tissues?
2. If we accept, that the measured fixed samples are fir such comparisons, how is related the elasticity variation of single sample (in several different areas of the sample) to those observed on global comparison (normal, benign, malign)? How many maps were effectuated per tissue sample? The presented 10x10 point area covering 30 by 30 micrometers would be good as representative but only toghether with a graph showing the variation within a single sample of at least ten different areas.
3. The authors name the Hertzian approach as the model for elasticity quantification, and as they use a sharp tip (10 nm radius) on a hard cantilever why do they expect to measure linear behavior on force curves (this would be tha case of a flat indenter)? Deviation from linear curve is the consequence of the used indenter, why do they use it as a marker for malignancy? This would require more clarification.
4. The Authors present on Figure 2 d-e modeled sizes for cells within tissue, were there any stainings performed to identify the origin of these claimed craters, they resemble very much to nuclei, both in size and shape, however some staining would clarify clearly.
Furthermore, at the caption of this picture the height distribution is given as 0.0, +/- 0.31um the deviation being larger than the mean value. Were there some flattens or other image manipulations involved?
5. Supplementary figure S3 shows the distribution of elastic modulus of glass substrate. How exaclty was this evaluated? The well established approaches use indentation data after subtraction of deflection of cantilever recorded on hard (typically glass) surface. Was the cantilever really stiff enough to indent the glass? Please clarify the used calibration and calculation methods.
6. Figures S6-7-8 present elastic moduli distibutions from different patients. Would this mean that one map / tissue sample was ment to characterize the sample? In some cases the gaussian peak numbers are not convincing. Depending on the bin size, one would expect different outcomes. How exactly the fittings were effectuated on elasticity distribution data? What crieteria was used to distinguish between single and multi-gaussian fittings? This question is cardinal, as the paper targets the multi-gaussian distributions as markers for malignancy. Please clarify.
Comments: please be more careful within the text not to confuse the term elasticity with the values of elastic moduli. If the elastic moduli are higher the elasticity is lower not counterwise.
Figure 3 g-i presents values of stiffness in N/m, while the histogram below at panel j-l is labelled as Elastic Modulus in MPa. Technically this is ok, but it would be easier for the reader to evaluate the information if both would have tha same type.
The authors state that force volume maps were aquire 3-4.5 um apart, but these would mean overlapping areas with map size of 30um.
Typo: number of patients is 32 in line 354 while the summarysing table shows 33.
Line 410: Cantilever speed was 1um/s for tissue and 0.3um/s for cells. What cells is the text about? There is not mentioned solely tissue measurements.
Author Response
Dear Editor,
We hereby resubmit our revised manuscript (Manuscript ID: diagnostics-1767451), entitled "Melanoma Detection by AFM Indentation of Histological Specimens", to your journal, Diagnostics. We have revised the manuscript according to the reviewers’ comments and suggestions. The revised text is colored in blue in the revised manuscript. The details of our revision are summarized below in response to each of the reviewers’ comments. We look forward to hearing from you regarding the revised manuscript.
We appreciate your time and effort in responding to our manuscript.
Yours Sincerely,
Wonseok Lee, Ph.D.
Department of Electrical Engineering
Korea National University of Transportation

Reviewer 2 Report
The authors explored the possibility of AFM for identifying differences between nevus and melanoma tissues. The results of indentation experiment measured by AFM clearly showed these differences. I don't think there is any particular problem with the content of the paper for publication, but I have some detailed comments regarding AFM measurement.
1. AFM topographic image
The AFM images in Figure 2a and 2b look like AFM images stretched in one direction, respectively. Was it an effect of a mechanical drift of AFM system, or the real surface structure of the tissue obtained from the human skin by chance?
2. AFM tip used for indentation experiment
The author used a conventional AFM cantilever with a sharp tip (tip radius < 10 nm) in indentation experiment. However, generally, a colloidal probe with a diameter of several um is often used to measure the elastic modulus of the human coenocyte. It is easier to approximate the Hertz model with the experimentally obtained force curve without huge variation in experimental data by using a relatively blunt tip. Why the author used a conventional AFM cantilever? Is it just because the conventional tip is easy to use or is it necessary to use a sharp probe to obtain quantitative data? If you have any reason for this, it's better to mention it in the main text.
3. Force curve measurement
In the indentation experiment, the author set the threshold value of the force curve measurement at about 2 uN, and the physical indentation depth was about 500 nm. Is this an optimized value for investigating each difference between normal, nevus and melanoma specimens? If you push harder from 2 uN, can you see a more significant difference?
Relation to this question, I would like to ask one question about sample preparation. The author noted that “the histo-logical specimen could withstand the micro-newton force because it was a fixed tissue with chemical reagent rather than raw biological tissue.” The standard histological procedures were used for the sample preparation in this research. However, I expect that it is better to use a living sample without chemical treatment to see the huge difference between normal, nevus and melanoma specimens. Is there any idea for improvement in the sample preparation without overly increasing the sample preparation procedure?
4. Optimal data size of AFM image
In order to use this method as a practical measurement tool, it is necessary to further speed up this method, but on the other hand, it is also necessary to maintain quantitative data acquisition. Is it possible to reduce the lateral pixel size and the physical scan distance of XYZ in the detection of each specimen?
Author Response

(The authors gave the same response as above.)
